# Large-Scale Oil Palm Trees Detection from High-Resolution Remote Sensing Images Using Deep Learning

**Hery Wibowo \*, Imas Sukaesih Sitanggang** **, Mushthofa Mushthofa and Hari Agung Adrianto**

Department of Computer Science, IPB University, Bogor 16680, Indonesia
\* Correspondence: pascasarjanaipbhery@apps.ipb.ac.id; Tel.: +62-821-8000-5654

**Abstract:** Tree counting is an important plantation practice for biological asset inventories, etc. The application of precision agriculture in counting oil palm trees can be implemented by detecting oil palm trees from aerial imagery. This research uses the deep learning approach using YOLOv3, YOLOv4, and YOLOv5m in detecting oil palm trees. The dataset consists of drone images of an oil palm plantation acquired using a Fixed Wing VTOL drone with a resolution of 5cm/pixel, covering an area of 730 ha labeled with an oil palm class of 56,614 labels. The test dataset covers an area of 180 ha with flat and hilly conditions with sparse, dense, and overlapping canopy and oil palm trees intersecting with other vegetations. Model testing using images from 24 regions, each of which covering 12 ha with up to 1000 trees (for a total of 17,343 oil palm trees), yielded F1-scores of 97.28%, 97.74%, and 94.94%, with an average detection time of 43 s, 45 s, and 21 s for models trained with YOLOv3, YOLOv4, and YOLOv5m, respectively. This result shows that the method is sufficiently accurate and efficient in detecting oil palm trees and has the potential to be implemented in commercial applications for plantation companies.

**Keywords:** deep learning; drone; oil palm; tree detection; YOLOv3; YOLOv4; YOLOv5



## 1. Introduction

Oil palm is an essential agricultural economic crop in many tropical countries such as Indonesia, Malaysia, Thailand, and Colombia. The primary use of oil palm is to produce palm oil, which is not only used to make vegetable oil but also as raw material for cosmetics, biodiesel, and others. Palm oil is the main source of vegetable oil due to its high yield compared to other vegetable oils [1], and it is the most consumed vegetable oil in the world [2]. Palm oil is the most important vegetable oil globally in production and trade [3].

Tree counting is an important plantation practice for biological asset inventories, fresh fruit bunch production estimation, fertilization and maintenance budgeting, plant growth/health monitoring, replanting, plant layout planning, etc. Counting trees manually is expensive, labor-intensive, and prone to errors. Most plantations are forced to estimate the amount of fresh fruit bunch production by multiplying the total area by the number of oil palm trees per hectare, which often results in significant inaccuracy due to the heterogeneity of the land surface, which is hilly and undulating, as well as the presence of rivers, wasteland, and forests. Remote sensing is a solution to these problems because of the broad view of the plantation area, and it is a means of counting oil palm trees [4].

Yin et al. [5] stated that drone remote sensing is cheaper and more flexible than satellite imagery on an industrial scale. The rapid development of drone technology, information technology, and sensor technology allows drones to also be applied to various fields of agriculture and forestry. Drone remote sensing technology can monitor large areas of plantations with high spatial resolution, so it is widely used for oil palm research such as tree counting [6–8], oil palm harvest prediction [9], plant nutrition monitoring [10], and plant health monitoring [11].

The use of remote sensing as an alternative to traditional methods has led many researchers to find various techniques and ways to increase the accuracy of counting oil palm trees. Oil palm plantations have a unique shape and pattern based on the discrimination of oil palms from non-oil palms using spectral analysis, texture analysis, edge enhancement, segmentation processes, morphological analysis, and blob analysis, as reported by Shafri et al. [12]. Syed Hanapi et al. [13] reviewed several methods for detecting and delineating trees in forests and oil palm plantations, including several sample algorithms from techniques such as image processing, machine learning, point cloud, and deep learning. There are still gaps for improvement and development, especially in the methods used. With the improvement of remote sensing technology, it is possible to focus on the practicality of the methods used at lower costs which yield results of a higher quality.

Currently, deep learning methods have been widely used in various applications, particularly in image detection and classification [14–19]. Furthermore, oil palm research using deep learning has been widely carried out, including tree counting [20], fruit ripeness classification [21], plant health classification [22], the mapping of oil palm land [23], and the counting of Fresh Fruit Bunches (FFB) [24]. Deep learning is a subset of machine learning belonging to the broader artificial intelligence family. Deep learning is based on an artificial neural network (ANN) with many hidden layer networks. Deep learning has a network capable of implementing supervised or unsupervised learning from labeled or unlabeled data [25].

YOLO (You Only Look Once) is a new approach in computer vision for object detection, namely, recognizing objects and their location in images or videos. YOLO uses a Convolutional Neural Network (CNN) architecture, applies a single neural network to the entire image, divides the image into grids, and predicts the coordinates and class probabilities of the bounding box [26]. The development of YOLOv1 was first initiated by Redmon et al. [26]. The following year, YOLOv2, or YOLO9000, was developed by Redmon and Farhadi [27], YOLOv3 was developed by Redmon and Farhadi [28], YOLOv4 was developed by Bochkovskiy et al. [29], and YOLOv5 was developed by Jocher [30]. Several oil palm studies using the YOLO algorithm include tree counting [31], Fresh Fruit Bunches (FFB) counting [32], and harvesting systems [33].

In recent years, there have been many deep learning studies for detecting and counting oil palm trees. Li et al. [34] used the Convolutional Neural Network (CNN) LeNet architecture and sliding window technique approach to detect oil palm trees from Quick-Bird high-resolution satellite imagery. This method achieves a detection accuracy of 96%. Li et al. [35] proposed Deep Convolutional Neural Network (DCNN) AlexNet architecture, a sliding window, and post-processing to detect large-scale oil palm trees from Quickbird high-resolution satellite imagery. The object under study is a dense and overlapping oil palm tree with various scenes of oil palm trees, backgrounds, vegetation, and settlements with an accuracy of 92–97%. Mubin et al. [36] proposed the CNN method of the LeNet architecture combined with GIS in processing and storing data, as well as high-resolution Worldview-3 satellite imagery to capture object images. The accuracy rates for detecting young and mature oil palm trees were 95.11% and 92.96%, respectively. Bonet et al. [37] used the Transfer Learning CNN approach with the VGG-16 architecture (without the last layer) for feature extraction. The SVM classifier produces 97-98% accuracy in detecting oil palm trees from UAV images. Liu et al. [38] proposed the Faster R-CNN method to build a model to detect and automatically count oil palm trees from UAV images. Data testing was carried out in three regions with accuracy rates of 97.06%, 96.58%, and 97.79%, respectively.

In this study, we propose the detection of oil palm trees from drone imagery over large areas. The research location consists of an oil palm plantation area with a flat and hilly topography. In a flat area, the distance between the plant canopy is sparse and close together, whereas, in a hilly area, the plant canopy distance overlaps when viewed from the drone image. In addition, oil palm tree leaf areas can also intersect with other vegetations with leaf colors similar to oil palm. Hilly conditions with overlapping canopies and intersecting with other vegetations are challenging for object detection algorithms to identify oil palm trees. The method used in this study is a deep learning approach based on

YOLOv3, YOLOv4, and YOLOv5m due to its real-time object detection capability, which typically has a higher accuracy in object detection and a faster computation time than other deep learning algorithms [28–30].

The rest of this paper is organized as follows. Section 2 presents the research plan, the study area and datasets, the data preprocessing and model building steps, and the evaluation metrics; Section 3 describes the training and testing results of our proposed method; Section 4 describes the performance and limitation methods; Section 5 presents some important conclusions of this research.

## 2. Materials and Methods

### 2.1. Overview

The research plan in this study is presented in the flowchart in Figure 1. Preprocessing the data on the drone images produces training data, validation data, and testing data. The training and validation process uses YOLO pre-trained weights for convolutional layers to be more accurate and to avoid lengthy model training steps. Hyperparameter tuning is also performed in order to find the optimal model during training, which will then be used during model testing. The detection results will be evaluated by comparing the accuracies and detection times of YOLOv3, YOLOv4, and YOLOv5m.

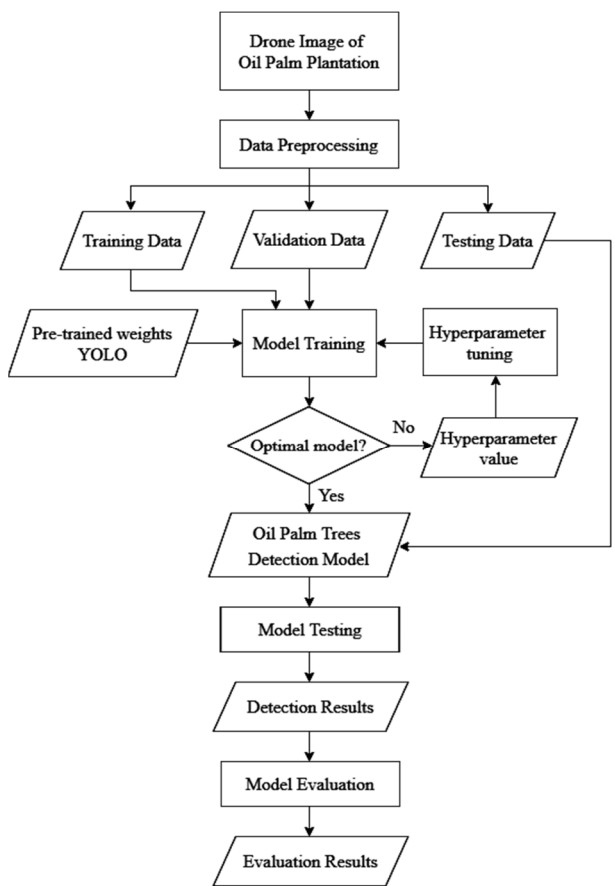

**Figure 1.** Research plan in this study.

### 2.2. Study Area and Datasets

The research location in this study is an oil palm plantation located in Jambi province, Indonesia, as shown in Figure 2. The image was acquired from 2–3 May 2021 using a Fixed Wing VTOL drone, shown in Figure 3, at an altitude of 200 m above ground level with a resolution of 5 cm/pixel. The drone specifications can be seen in Table 1. Each of the sample dataset areas (training, validation, testing) includes oil palm trees under the criteria of young plants (the planting year 2013–2014) and mature plants (the planting year

2009–2011) in oil palm plantation areas with flat and hilly contours with sparse, dense, and overlapping canopy spacing conditions, as well as oil palm trees intersecting with other vegetations. The training and validation area covers 730 ha, while the testing area covers 180 ha. The distribution of datasets based on the blocks area in regions in oil palm plantations can be seen in Table 2.

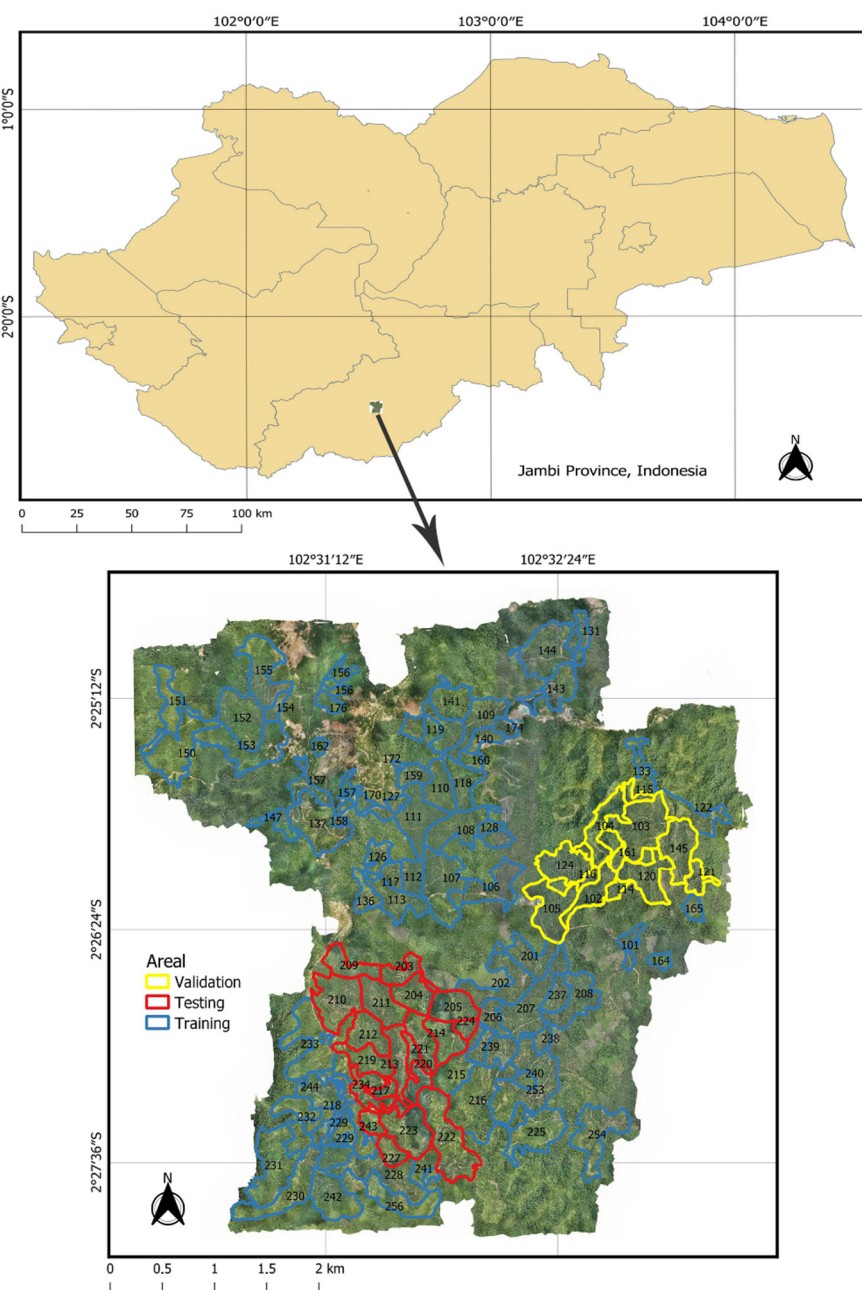

**Figure 2.** The study area is located in Jambi province, Indonesia. The drone images of oil palm plantations consist of three areas. Blue and yellow polygons are for training and validation areas, while red polygons are for testing areas.

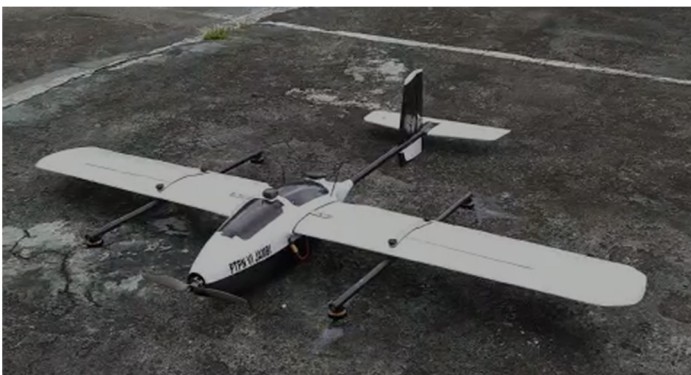

**Figure 3.** Fixed Wing VTOL drone.

**Table 1.** Drone specifications.

| Attributes | Description |
|---|---|
| Wingspan | 2000 mm |
| Weight | 3.8 kg |
| Radio control | 2.4 GHz |
| Camera | RGB 24 mp support PPK GNSS |
| Telemetry type long-range | 15 km |
| Flying ability | Full system auto/fly without remote |
| Cruising flight maximum | 70 min/60 km |

**Table 2.** The distribution of datasets based on blocks area.

| Datasets | Blocks |
|---|---|
| Training | 101,106,107,108,109,110,111,112,113,117,118,119,122,126,127,128,131,133,136, 137,140,141,143,144,147,150,151,152,153,154,155,156,157,158,159,160,162,164, 165,170,172,174,176,201,202,206,207,208,215,216,218,225,228,229,230,231,232, 233,237,238,239,240,241,242,243,244,253,254,256. |
| Validation | 102,103,104,105,114,115,116,120,121,124,145,161. |
| Testing | 203,204,205,209,210,211,212,213,214,217,219,220,221,222,223,224,227,234,257. |

*2.3. Data Preprocessing*

Before performing input data processing in the YOLO architecture, we first preprocess the data with the following steps:

1.  Cropping the drone images for the training (311 images) and validation (66 images) datasets into grids with a size of 3943 × 3943 pixels, which corresponds to 200 m × 200 m (4 ha) using QGIS. An example of the image can be seen in Figure 4.
2.  Manually identifying and labeling 56,614 oil palm trees in the training and validation datasets using LabelImg. The composition of the training data is 80% (45,290), and that of the validation data is 20% (11,324). An example of image labeling can be seen in Figure 5.
3.  Cropping 24 drone images on the testing data block into grids with an image size of 7886 × 5914 pixels, which corresponds to 400 m × 300 m (12 ha) using QGIS. An example of the image can be seen in Figure 6.

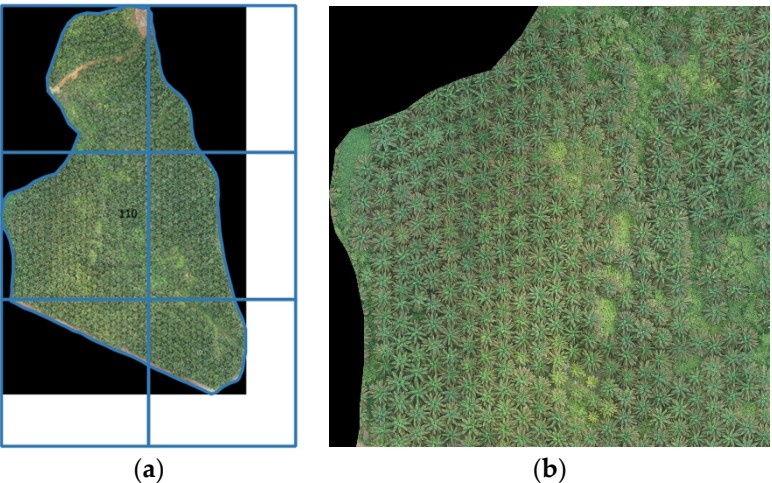

**Figure 4.** An example of image cropping 3943 × 3943 pixels: (**a**) Mapping on block area; (**b**) Cropping results.

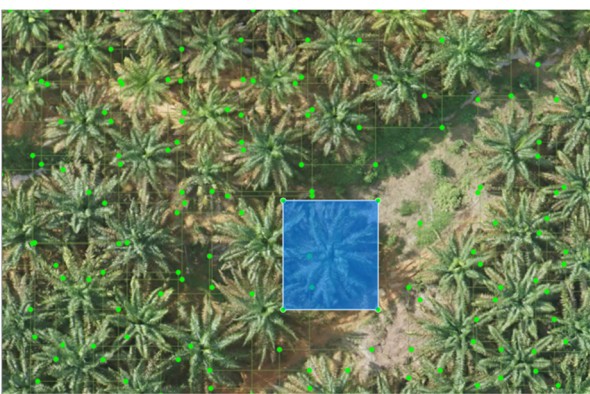

**Figure 5.** An example of image labeling. Each oil palm tree object is labeled using a bounding box.

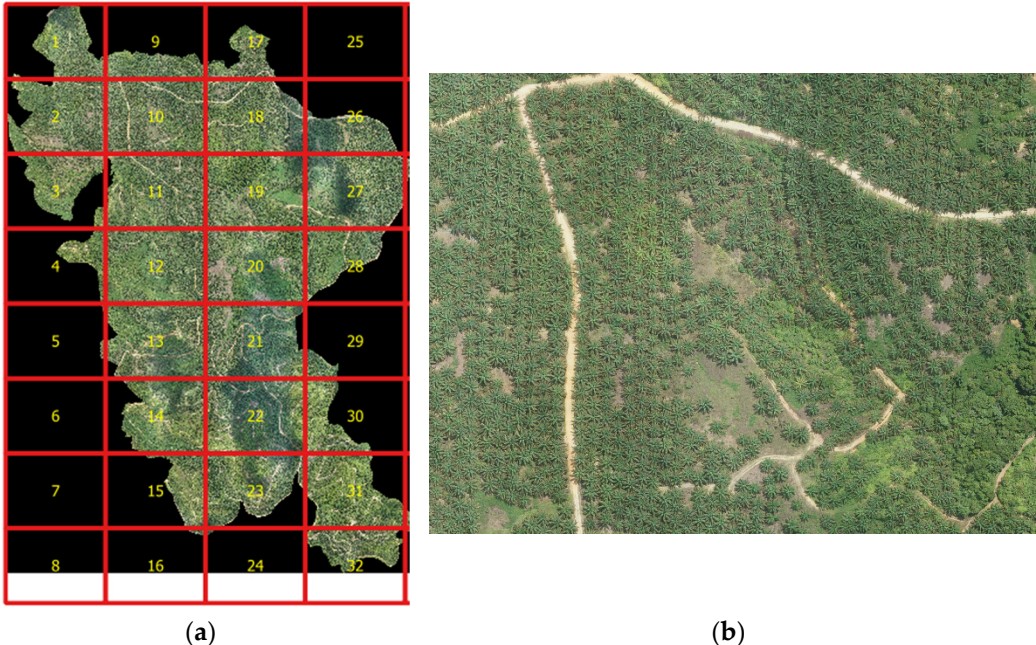

**Figure 6.** An example of image cropping 7886 × 5914 pixels: (**a**) Mapping on block area; (**b**) Cropping results on grid number 10. There are 24 testing areas from cropping results on the map, namely on grid numbers 1,2,3,9,10,11,12,13,14,15,17,18,19,20,21,22,23,26,27,28,29,30,31,32.

### 2.4. Model Development

The model development uses the Darknet framework on YOLOv3 [28] and YOLOv4 [29], while YOLOv5m [30] uses the PyTorch framework. We tried several input sizes of the network (width × height)—416 × 416, 608 × 608, 832 × 832, and 1024 × 1024 for YOLOv3, YOLOv4, and YOLOv5m—to obtain the best alternative model. The hyperparameters used are batch size, subdivision, momentum, decay, and learning rate. The larger the network input size, the greater the computational process. Therefore, some adjustments to the batch size and subdivision values were made, while the momentum, decay, and learning rate were set to their default values. The network input size and hyperparameter scenarios for YOLOv3, YOLOv4, and YOLOv5m are given, respectively, in Tables 3–5. We used YOLO pre-trained weights for convolutional layers to increase the accuracy and to avoid longer training times. Hyperparameter tunings were also performed to optimize the training process.

**Table 3.** YOLOv3 network input sizes and hyperparameters scenario for training and validation.

| Network Input Size (Width × Height) | Hyperparameters | | | | |
|---|---|---|---|---|---|
| | Batch | Subdivision | Momentum | Decay | Learning Rate |
| 416 × 416 | 64 | 16 | 0.9 | 0.0005 | 0.001 |
| 608 × 608 | 64 | 16 | 0.9 | 0.0005 | 0.001 |
| 832 × 832 | 64 | 16 | 0.9 | 0.0005 | 0.001 |
| 1024 × 1024 | 64 | 32 | 0.9 | 0.0005 | 0.001 |

**Table 4.** YOLOv4 network input sizes and hyperparameters scenario for training and validation.

| Network Input Size (Width × Height) | Hyperparameters | | | | |
|---|---|---|---|---|---|
| | Batch | Subdivision | Momentum | Decay | Learning Rate |
| 416 × 416 | 64 | 16 | 0.949 | 0.0005 | 0.001 |
| 608 × 608 | 64 | 16 | 0.949 | 0.0005 | 0.001 |
| 832 × 832 | 64 | 32 | 0.949 | 0.0005 | 0.001 |
| 1024 × 1024 | 64 | 64 | 0.949 | 0.0005 | 0.001 |

**Table 5.** YOLOv5m network input sizes and hyperparameters scenario for training and validation.

| Network Input Size (Width × Height) | Hyperparameters | | | |
|---|---|---|---|---|
| | Batch | Momentum | Decay | Learning Rate |
| 416 × 416 | 64 | 0.937 | 0.0005 | 0.01 |
| 608 × 608 | 32 | 0.937 | 0.0005 | 0.01 |
| 832 × 832 | 16 | 0.937 | 0.0005 | 0.01 |
| 1024 × 1024 | 8 | 0.937 | 0.0005 | 0.01 |

### 2.5. Evaluation Metrics

The measurement of model performance in this study used Recall, Precision, and F1-score [39], as shown in Equations (1)–(3). Recall measures how well the model can detect oil palm trees, precision measures how accurately the model predicts oil palm trees, and F1-score is the harmonic mean of recall and precision. In addition, detection time is also used as a metric in model evaluation in order to be able to compare model efficiency during detection. Average IoU is also needed to assess the accuracy of the bounding box location for detection [40], as shown in Equation (4); the illustration can be seen in Figure 7.

$$\text{Recall} = \frac{\text{TP}}{(\text{TP} + \text{FN})} \tag{1}$$

$$\text{Precision} = \frac{\text{TP}}{(\text{TP} + \text{FP})} \tag{2}$$

$$F1 - score\ = \frac{(2 \times\ Recall\ \times\ Precision)}{(Recall\ +\ Precision)} \tag{3}$$

TP (True Positive) = The objects of oil palm trees were detected as oil palm trees.
FP (False Positive) = Objects other than oil palm trees were detected as oil palm trees.
FN (False Negative) = The objects of oil palm trees were not detected as oil palm trees.

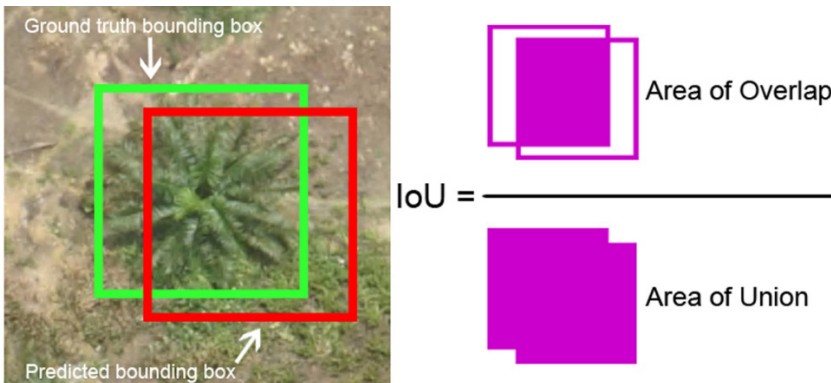

**Figure 7.** Illustration of IoU (Intersect over Union). The ground truth bounding box from the manual labeling of validation data determines where our object is in the image, while the predicted bounding box is from the trained model.

$$IoU = \frac{Area\ of\ Overlap}{Area\ of\ Union} \tag{4}$$

## 3. Results

### 3.1. Training Results

Training and validation with four network input size scenarios in YOLOv3 were carried out for 8000 iterations with a saved model every 1000 iterations, while the training and validation of YOLOv4 and YOLOv5m were carried out for 6000 iterations with a saved model every 500 iterations. Each saved model iteration is validated with a threshold variation of 0.1, 0.2, 0.3, 0.4, 0.5, 0.6, 0.7, 0.8, and 0.9 for the Precision, Recall, F1-score, and Average IoU values. The selection of the threshold of each saved model iteration prioritizes the Recall value—in this case, the model's ability to detect oil palm tree objects. The determination of the best model as a result of training from the saved model iteration is based on the highest Precision, Recall, and F1-score values for YOLOv3 and YOLOv4, while YOLOv5m is based on 6000 iterations because the training tends to be stable overall. The results of the training evaluation for each network input size for YOLOv3, YOLOv4, and YOLOv5m can be seen in Supplementary Spreadsheet S1. The best models obtained from each network input size on YOLOv3, YOLOv4, and YOLOv5m are shown in Tables 6–8, respectively. These models will be used for testing the test data.

**Table 6.** The best model for each network input size on YOLOv3 is based on training and validation.

| Network Input Size (Width × Height) | Iteration | Threshold | Precision | Recall | F1-Score | Average IoU (%) |
|---|---|---|---|---|---|---|
| 416 × 416 | 4000 | 0.2 | 0.84 | 0.99 | 0.91 | 72.71 |
| 608 × 608 | 7000 | 0.2 | 0.85 | 0.99 | 0.91 | 74.84 |
| 832 × 832 | 7000 | 0.4 | 0.86 | 0.99 | 0.92 | 76.52 |
| 1024 × 1024 | 8000 | 0.4 | 0.86 | 0.99 | 0.92 | 76.42 |

**Table 7.** The best model for each network input size on YOLOv4 is based on training and validation.

| Network Input Size (Width × Height) | Iteration | Threshold | Precision | Recall | F1-Score | Average IoU (%) |
|---|---|---|---|---|---|---|
| 416 × 416 | 5000 | 0.3 | 0.85 | 0.99 | 0.91 | 73.92 |
| 608 × 608 | 3500 | 0.4 | 0.86 | 0.99 | 0.92 | 76.17 |
| 832 × 832 | 4000 | 0.4 | 0.86 | 0.99 | 0.92 | 76.76 |
| 1024 × 1024 | 2000 | 0.4 | 0.86 | 0.99 | 0.92 | 70.35 |

**Table 8.** The best model for each network input size on YOLOv5m is based on training and validation.

| Network Input Size (Width × Height) | Iteration | Threshold | Precision | Recall | F1-Score | Average IoU (%) |
|---|---|---|---|---|---|---|
| 416 × 416 | 6000 | 0.5 | 0.97 | 0.89 | 0.93 | 71.20 |
| 608 × 608 | 6000 | 0.5 | 0.97 | 0.89 | 0.93 | 73.60 |
| 832 × 832 | 6000 | 0.5 | 0.97 | 0.89 | 0.93 | 74.70 |
| 1024 × 1024 | 6000 | 0.5 | 0.97 | 0.89 | 0.93 | 75.10 |

*3.2. Testing Results*

The oil palm detection test is carried out on the testing data block with image grids measuring 7886 × 5914 pixels, which corresponds to 400 m × 300 m (12 ha), with as many as 24 images (regions) and as many as 17,343 oil palm trees objects (ground truth), as shown in Table 9. The oil palm tree detection test results using the models from YOLOv3, YOLOv4, and YOLOv5m, referred to in Tables 6–8, are given in Tables 10–12, respectively. The testing results per region based on the network input size for YOLOv3, YOLOv4, and YOLOv5m can be seen in Supplementary Spreadsheet S2.

**Table 9.** Oil palm tree detection testing area.

| Region | Ground Truth |
|---|---|
| Region 1 (grid 1) | 547 |
| Region 2 (grid 2) | 1079 |
| Region 3 (grid 3) | 324 |
| Region 4 (grid 9) | 423 |
| Region 5 (grid 10) | 1301 |
| Region 6 (grid 11) | 1213 |
| Region 7 (grid 12) | 991 |
| Region 8 (grid 13) | 982 |
| Region 9 (grid 14) | 754 |
| Region 10 (grid 15) | 407 |
| Region 11 (grid 17) | 251 |
| Region 12 (grid 18) | 1117 |
| Region 13 (grid 19) | 731 |
| Region 14 (grid 20) | 674 |
| Region 15 (grid 21) | 1170 |
| Region 16 (grid 22) | 1090 |
| Region 17 (grid 23) | 889 |
| Region 18 (grid 26) | 539 |
| Region 19 (grid 27) | 1209 |
| Region 20 (grid 28) | 440 |
| Region 21 (grid 29) | 98 |
| Region 22 (grid 30) | 190 |
| Region 23 (grid 31) | 485 |
| Region 24 (grid 32) | 439 |

**Table 10.** YOLOv3 model evaluation results.

| Network Input Size (Width × Height) | GT | TP | FP | FN | Recall (%) | Precision (%) | F1-Score (%) | Detection Time (s) |
|---|---|---|---|---|---|---|---|---|
| 416 × 416 | 17,343 | 15,542 | 235 | 1801 | 89.62 | 98.51 | 93.85 | 42 |
| 608 × 608 | 17,343 | 16,403 | 146 | 940 | 94.58 | 99.12 | 96.80 | 41 |
| 832 × 832 | 17,343 | 16,073 | 19 | 1270 | 92.68 | 99.88 | 96.14 | 43 |
| 1024 × 1024 | 17,343 | 16,432 | 7 | 911 | 94.75 | 99.96 | 97.28 | 43 |

**Table 11.** YOLOv4 model evaluation results.

| Network Input Size (Width × Height) | GT | TP | FP | FN | Recall (%) | Precision (%) | F1-Score (%) | Detection Time (s) |
|---|---|---|---|---|---|---|---|---|
| 416 × 416 | 17,343 | 14,946 | 145 | 2397 | 86.18 | 99.04 | 92.16 | 42 |
| 608 × 608 | 17,343 | 15,509 | 33 | 1834 | 89.43 | 99.79 | 94.32 | 45 |
| 832 × 832 | 17,343 | 16,600 | 23 | 743 | 95.72 | 99.86 | 97.74 | 45 |
| 1024 × 1024 | 17,343 | 10,699 | 7 | 6644 | 61.69 | 99.93 | 76.29 | 44 |

**Table 12.** YOLOv5m model evaluation results.

| Network Input Size (Width × Height) | GT | TP | FP | FN | Recall (%) | Precision (%) | F1-Score (%) | Detection Time (s) |
|---|---|---|---|---|---|---|---|---|
| 416 × 416 | 17,343 | 6935 | 69 | 10,408 | 39.99 | 99.01 | 56.97 | 20 |
| 608 × 608 | 17,343 | 15,257 | 84 | 2086 | 87.97 | 99.45 | 93.36 | 20 |
| 832 × 832 | 17,343 | 15,599 | 68 | 1744 | 89.94 | 99.57 | 94.51 | 20 |
| 1024 × 1024 | 17,343 | 15,717 | 49 | 1626 | 90.62 | 99.69 | 94.94 | 21 |

The comparison based on the evaluation results of the YOLOv3, YOLOv4, and YOLOv5m models refers to Tables 10–12. The object detection tests on 17,343 oil palm trees in 24 images (regions), in general, showed satisfactory results on YOLOv3, YOLOv4, and YOLOv5m. The precision values reached above 98%, as shown in Figure 8. This shows that the three models are sufficiently good at predicting the oil palm class correctly, with minor incorrect detections (false positives).

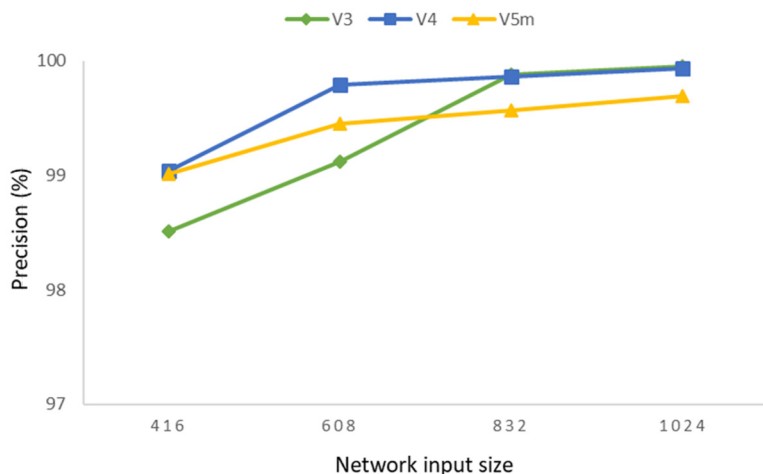

**Figure 8.** Comparison of the precision values of YOLOv3, YOLOv4, and YOLOv5m.

The recall value on YOLOv3, in general, is above 90%, indicating that the model is sufficiently good at detecting oil palm tree objects, with the highest value of 94.75% at the network input size 1024 × 1024. The highest recall value of YOLOv4 on the network input size 832 × 832 is 95.72%, but the network input size 1024 × 1024 only yields 61.69%. This is

likely due to the limited computing resources, causing poor training results when using the subdivision parameter value of 64. Based on the experimental results, where a larger network input size requires greater computational resources, we used larger subdivision parameter values for YOLOv3 and YOLOv4, with the consequence of reduced accuracies for both. The recall value on YOLOv5m shows an upward trend for the network input sizes $416 \times 416$, $608 \times 608$, $832 \times 832$, and $1024 \times 1024$, but there is a large gap between the network input size $416 \times 416$ and the others. This happens because the default value of the YOLOv5m network input size is $640 \times 640$; so, for a smaller network input size, the accuracy is lower. The comparison of the recall values of YOLOv3, YOLOv4, and YOLOv5m is shown in Figure 9.

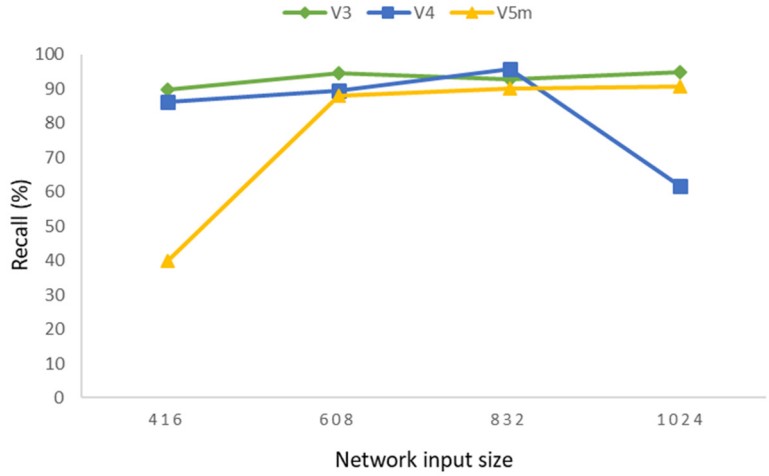

**Figure 9.** Comparison of the recall values of YOLOv3, YOLOv4, and YOLOv5m.

The poor training on the network input size $1024 \times 1024$ YOLOv4 is shown in Table S2 Supplementary Spreadsheet S1; there is a downward trend in the precision, F1-score, and average IoU values in the 2500 to 6000 iterations. This can also be seen in the comparison of the average IoU values in the training validation, as shown in Figure 10. YOLOv3 and YOLOv5m have an upward trend for the network input sizes $416 \times 416$, $608 \times 608$, $832 \times 832$, and $1024 \times 1024$, while YOLOv4 has an uptrend for the network input sizes $416 \times 416$, $608 \times 608$, and $832 \times 832$ but a downward trend at $1024 \times 1024$.

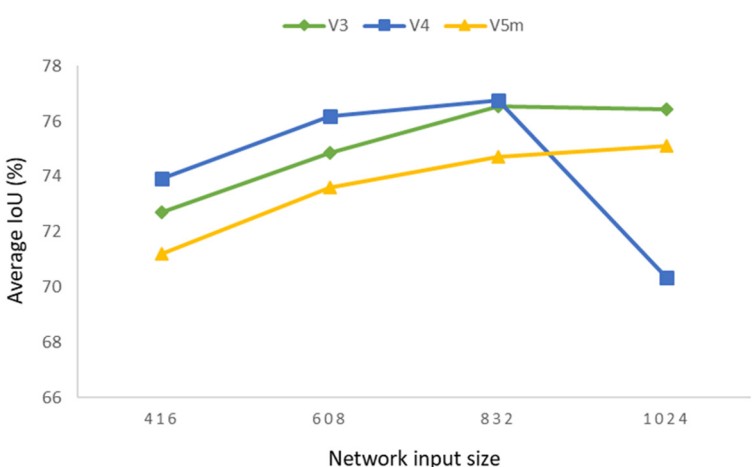

**Figure 10.** Comparison of the average IoU values of YOLOv3, YOLOv4, and YOLOv5m.

This is also in line with the F1-score values. YOLOv4 has an upward trend with values above 90% for the network input sizes $416 \times 416$, $608 \times 608$, and $832 \times 832$ but a downward trend in the network input size $1024 \times 1024$, with a value of 76.29%. YOLOv3

has a stable value at over 90% for the network input sizes 416 × 416, 608 × 608, 832 × 832, and 1024 × 1024, and YOLOv5m has an uptrend for the network input sizes 416 × 416, 608 × 608, 832 × 832, and 1024 × 1024. The low value of the F1-score on the network input sizes 1024 × 1024 YOLOv4 and 416 × 416 YOLOv5m is due to the low recall value. The comparison of the F1-score values of YOLOv3, YOLOv4, and YOLOv5m can be seen in Figure 11.

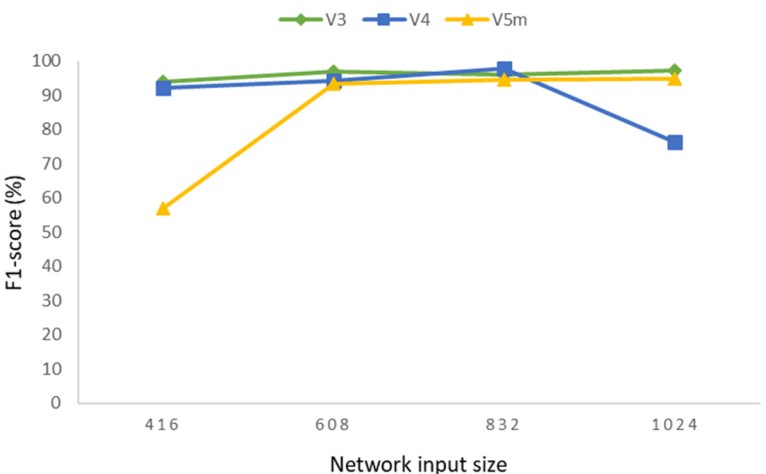

**Figure 11.** Comparison of the F1-score values of YOLOv3, YOLOv4, and YOLOv5m.

The average detection time in testing 24 images (regions) with a size of 7886 × 5914 pixels can be seen in Figure 12. On the network input sizes 416 × 416, 608 × 608, 832 × 832, and 1024 × 1024, YOLOv3 reaches 41–43 s, YOLOv4 reaches 42–45 s, and YOLOv5m reaches 20–21 s. There was no significant difference in the average detection time of YOLOv3 and YOLOv4, but YOLOv5m was twice as fast as YOLOv3 and YOLOv4.

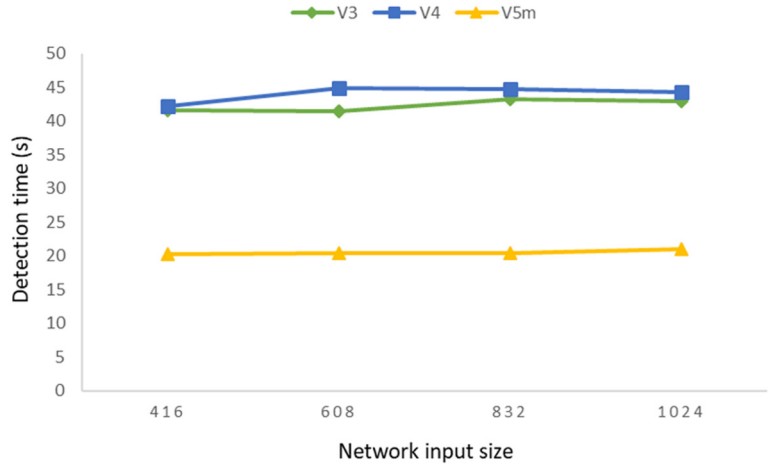

**Figure 12.** Comparison of the average detection time of YOLOv3, YOLOv4, and YOLOv5m.

Based on the experiments and visual detection results, we concluded that determining the best network input size model from each of the YOLOv3, YOLOv4, and YOLOv5m models is not based only on the criteria for the F1-score value and the detection time but also on the average IoU value. The F1-score is used to determine the overall accuracy, the detection time is used to find out how fast the model detects objects, and the average IoU is used to assess the accuracy of the bounding box location for detection. The accuracy of the bounding box location is important because the number of oil palm trees in one image file is quite large—up to hundreds to thousands of oil palm trees. The higher the IoU average

value, the better the accuracy of the location bounding box to the detected object, making for better visuals for large amounts of data, as shown in the comparison in Figure 13.

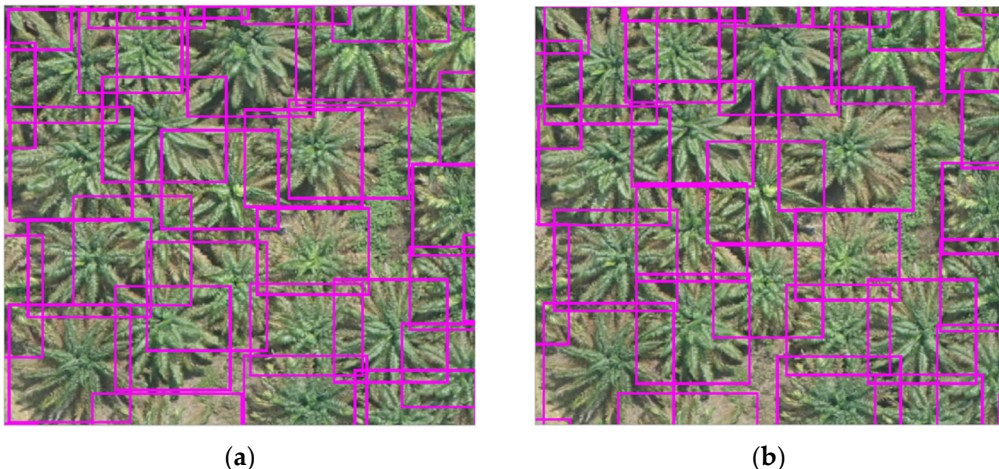

(**a**)  (**b**)

**Figure 13.** Comparison of the bounding box location to the detected object of the YOLOv3 detection results: (**a**) Poor accuracy on the network input size 416 × 416; (**b**) Good accuracy on the network input size 1024 × 1024.

The best model for the network input size on YOLOv3 is 1024 × 1024, with an F1-score of 97.28%, a detection time of 43 s, and an average IoU of 76.42%. On YOLOv4, the best model for the network input size is 832 × 832, with an F1-score of 97.74%, a detection time of 45 s, and an average IoU of 76.76%. On YOLOv5m, the best model for the network input size is 1024 × 1024, with an F1-score of 94.94%, a detection time of 21 s, and an average IoU of 75.1%. The image detection results per region based on the best model for YOLOv3, YOLOv4, and YOLOv5m can be seen in Supplementary Spreadsheet S3.

## 4. Discussion

YOLOv3, YOLOv4, and YOLOv5m have high accuracy and are fast in detecting oil palm trees (purple boxes), with only a few palm trees not detected (red circles). The models can detect oil palm trees with sparse and dense canopy conditions, as shown in Figures 14 and 15; besides that, the models are also able to detect oil palm trees with a hilly topography when viewed from the drone imagery of overlapping oil palm trees canopy, as shown in Figure 16. YOLOv5m was quite good at detecting oil palm trees in areas with hundreds of trees, but for the number of trees reaching thousands, there were many false negatives—namely, oil palm trees cannot be detected. YOLOv3 and YOLOv4 were quite good at detecting oil palm trees in areas with hundreds or even thousands of trees.

There were several shortcomings of the models during testing. It was quite challenging to detect oil palm trees with unhealthy oil palm conditions (stressed growth or nutrient deficiency), as shown by the red circles in Figures 17 and 18. In addition, the models had difficulty detecting (red circles) or incorrectly detecting (blue circles) for the condition where the oil palm trees intersect with other vegetations, so it is disguised, as shown in Figure 19. This weakness occurs because, at the time of labeling the oil palm class dataset, the majority of oil palm tree conditions are healthy/normal, while, for the unhealthy oil palm tree conditions and the oil palm trees intersecting with other vegetations (disguised), the labeling samples are few, so it is quite difficult to detect when testing the data.

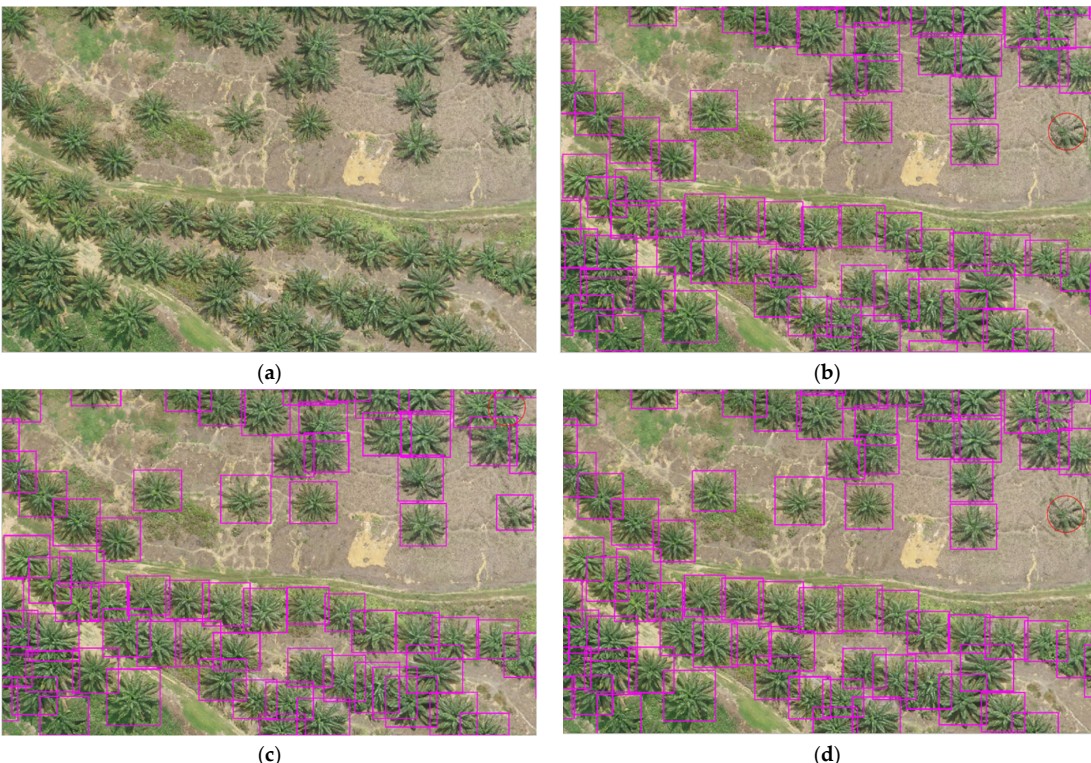

**Figure 14.** The sparse canopy condition of the oil palm trees: (**a**) Original image; (**b**) YOLOv3 detection; (**c**) YOLOv4 detection; (**d**) YOLOv5m detection.

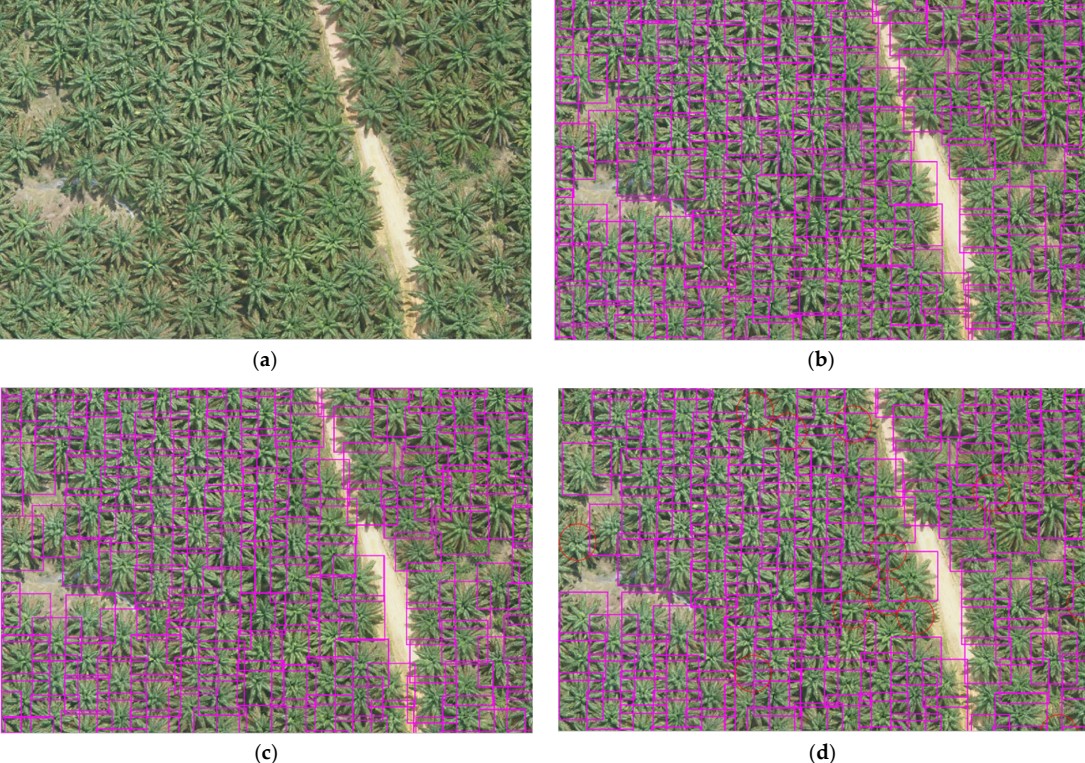

**Figure 15.** The dense canopy condition of the oil palm trees: (**a**) Original image; (**b**) YOLOv3 detection; (**c**) YOLOv4 detection; (**d**) YOLOv5m detection.

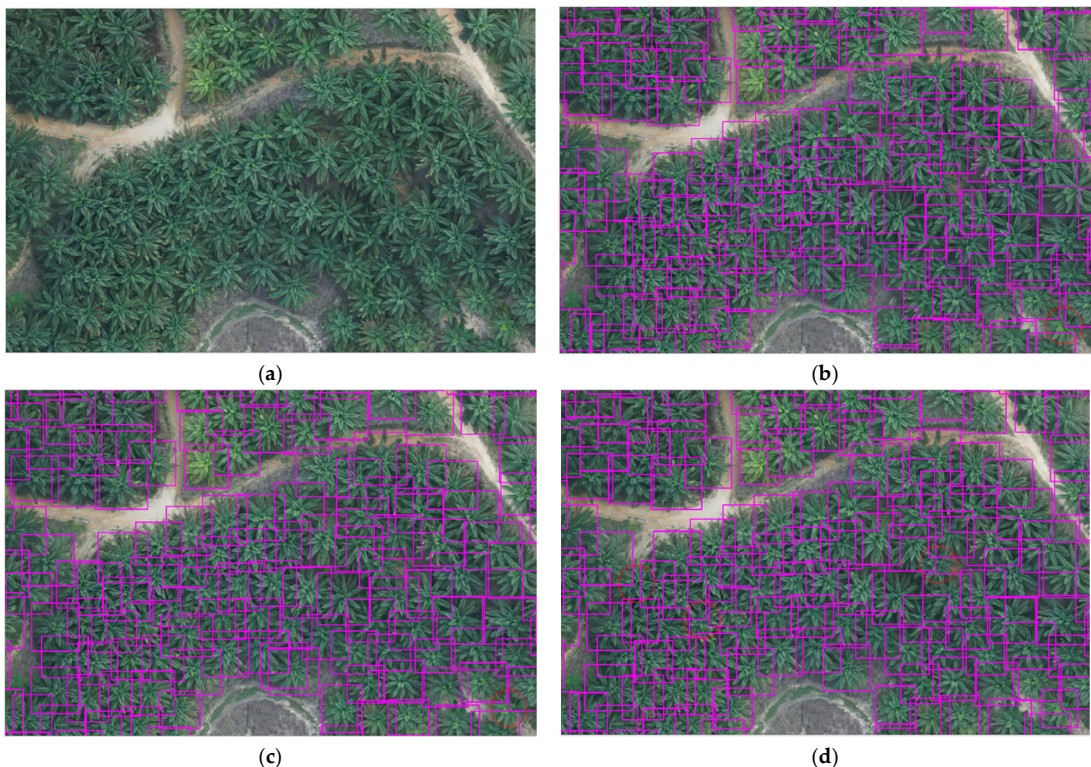

**Figure 16.** The overlapping canopy condition of the oil palm trees: (**a**) Original image; (**b**) YOLOv3 detection; (**c**) YOLOv4 detection; (**d**) YOLOv5m detection.

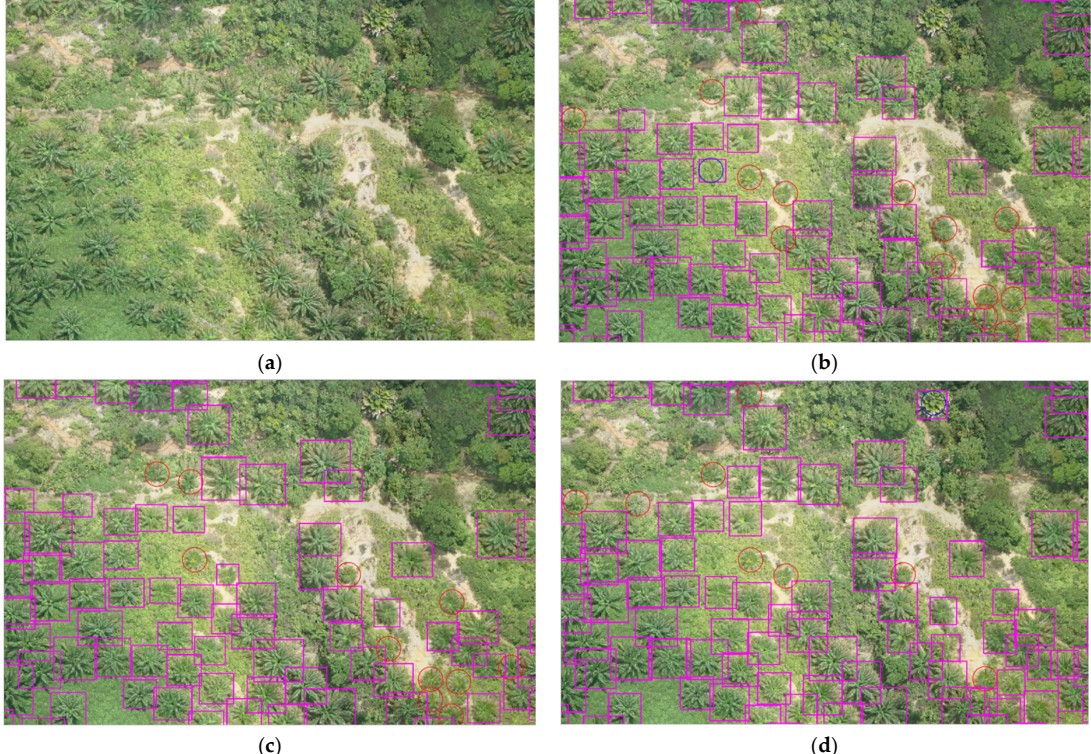

**Figure 17.** Oil palm trees with stressed growth: (**a**) Original image; (**b**) YOLOv3 detection; (**c**) YOLOv4 detection; (**d**) YOLOv5m detection.

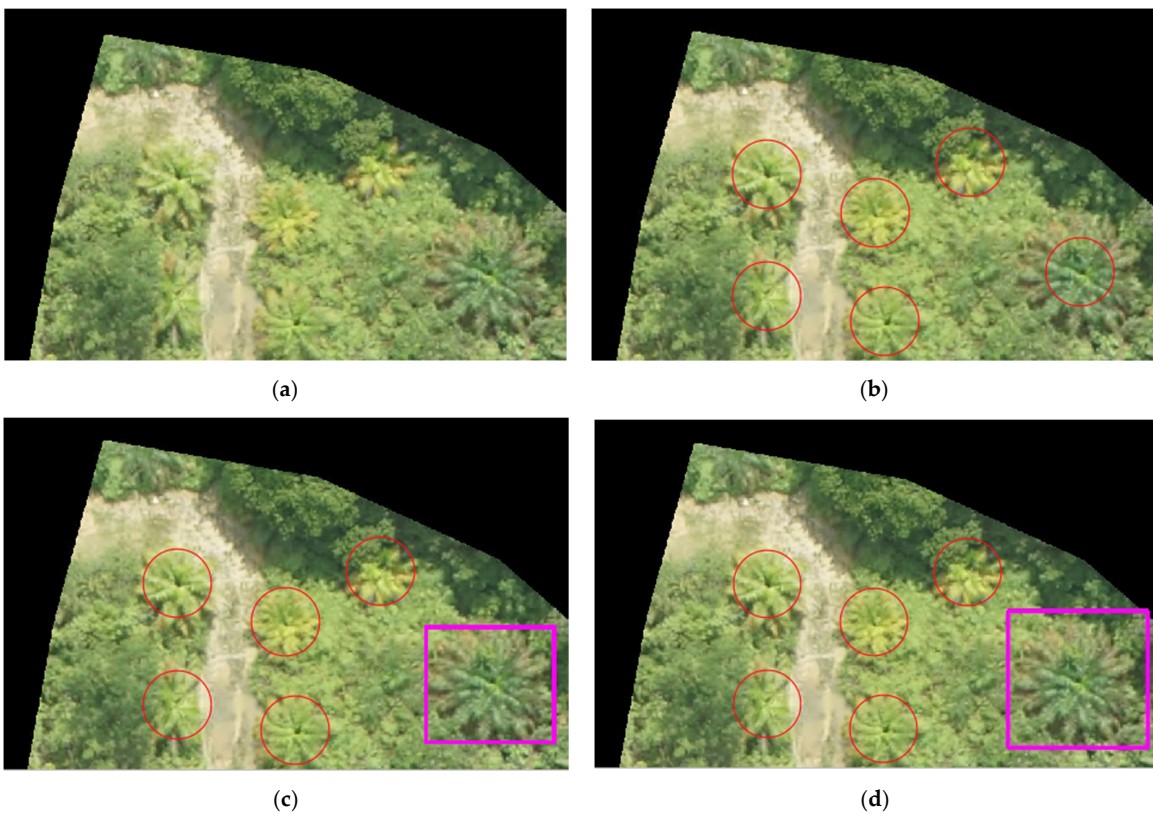

**Figure 18.** Oil palm trees with nutrient deficiency; (**a**) Original image; (**b**) YOLOv3 detection; (**c**) YOLOv4 detection; (**d**) YOLOv5m detection.

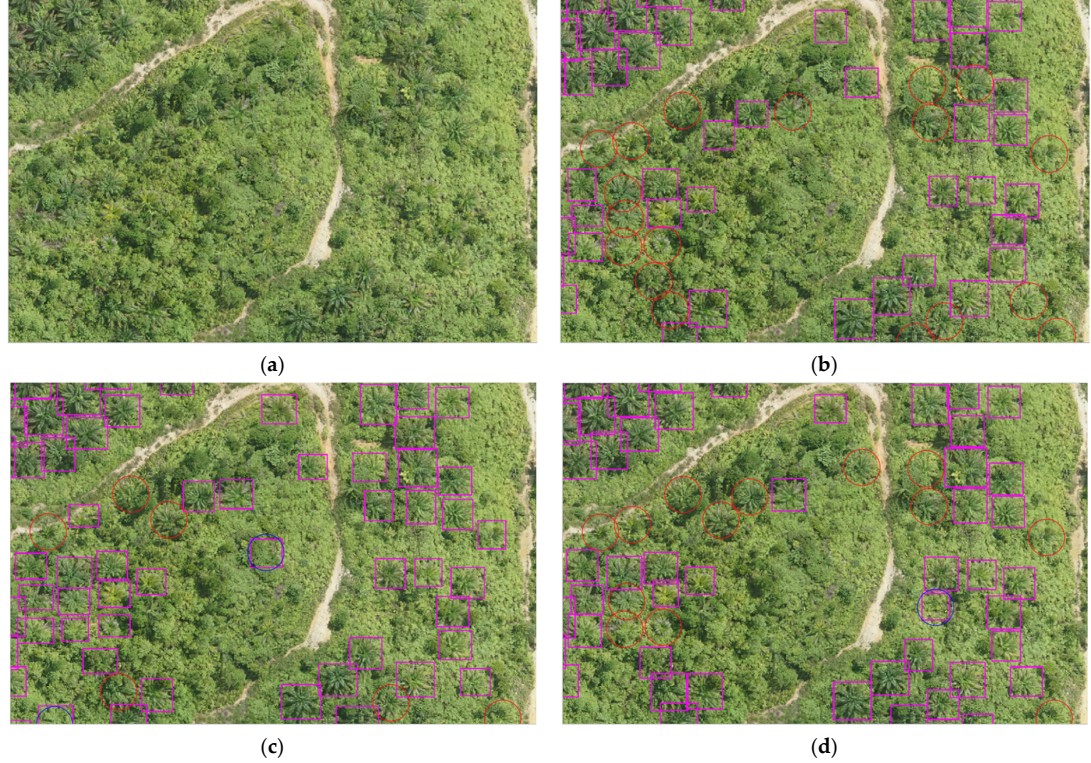

**Figure 19.** Oil palm trees camouflaged by other vegetations: (**a**) Original image; (**b**) YOLOv3 detection; (**c**) YOLOv4 detection; (**d**) YOLOv5m detection.

The three models can be applied to oil palm tree inventories in plantation companies because of their high accuracy, reaching 94–97%, while the fast detection time helps make the work more efficient for large-scale oil palm fields. Further development can add datasets of unhealthy oil palm trees (stressed growth/nutrient deficiency) and oil palm trees that are covered/camouflaged by other vegetations so that the model can recognize these objects during testing/detection. These datasets are not only used for an inventory of oil palm trees but can also detect/differentiate classes of healthy and unhealthy oil palm trees. Developing models with a larger network input size using hardware with higher resources is expected to increase accuracy. It is also possible to develop the model into a desktop/web-based application to make it easier for end-users to operate.

Compared to the previous oil palm tree counting research using deep learning [31,34–38], our method also provides accurate results, but what is different is that we present the results of the detection time on testing the test data and the accuracy of the bounding box. This information is important for the implementation of the large-scale detection of oil palm trees in areas reaching thousands to tens of thousands of hectares of oil palm land. With this large oil palm area, it requires a fast model in the detection and also needs the accuracy of the location of the bounding box for detection objects with a lot of data.

## 5. Conclusions

In this research, we have proposed an oil palm trees detection model using YOLOv3, YOLOv4, and YOLOv5m. The results of testing the YOLOv3, YOLOv4, and YOLOv5m models in four scenarios of the network input sizes $416 \times 416$, $608 \times 608$, $832 \times 832$, and $1024 \times 1024$ obtained the best model for YOLOv3 and YOLOv5m using the network input size of $1024 \times 1024$, while for YOLOv4, the best model used the network input size of $832 \times 832$. The test was carried out on 24 images/regions of 17,343 oil palm trees, with an average detection time of 43 s for YOLOv3, 45 s for YOLOv4, and 21 s for YOLOv5m. YOLOv3 obtained an F1 score of 97.28% and an average IoU of 76.42%, YOLOv4 obtained an F1 score of 97.74% and an average IoU of 76.76%, and YOLOV5m obtained an F1 score of 94.94% and an average IoU 75.1%. In terms of large-scale oil palm tree detection, accuracy is the top priority, so the recommendation for the best model is YOLOv3 and YOLOv4 because they have the highest accuracy, with a time difference of just under 25 s compared to YOLOv5m.

**Supplementary Materials:** Available online at https://ipb.link/supplementaryfiles. Spreadsheet S1: Results of training and validation on YOLOv3, YOLOv4, and YOLOv5m. Spreadsheet S2: Model evaluation per region on YOLOv3, YOLOv4, and YOLOv5m. Spreadsheet S3: Image detection results per region of the best models of YOLOv3, YOLOv4, and YOLOv5m.

**Author Contributions:** H.W. conducted the experiments and analysis and wrote the code and the article; I.S.S. designed the project, advised on research activities, guided in model building, and revised the article; M.M. advised on research activities, guided in model building, and revised the article, H.A.A. advised on research activities and revised the article. All authors have read and agreed to the published version of the manuscript.

**Funding:** This research and the APC was funded by IPB University. Grant number: 2887/IT3.L1/PT.01.03/M/T/2022.

**Institutional Review Board Statement:** Not applicable.

**Informed Consent Statement:** Not applicable.

**Data Availability Statement:** Not applicable.

**Acknowledgments:** The authors would like to thank IPB University for funding this research and the APC, and also PT Perkebunan Nusantara VI which is an Indonesian State-Owned Enterprise in the plantation sector as a provider of drones image datasets.

**Conflicts of Interest:** The authors declare no conflict of interest.

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
