# Peer review of "Large-Scale Oil Palm Trees Detection from High-Resolution Remote Sensing Images Using Deep Learning"

_2504-2289, doi:10.3390/bdcc6030089_

Round 1
Reviewer 1 Report
The manuscript focuses on the detection of oil palm trees from drone imagery over large areas. The obtained model evaluation results were high. It may be very useful in practice.
However, the results are not comprehensively described and discussed.
Section 3. Results should be supplemented with a more detailed description of the results.
Moreover, there is no discussion of the obtained results against the background of the available literature.
Author Response
Response to Reviewer 1 Comments
Point 1: The results are not comprehensively described and discussed. Section 3. Results should be supplemented with a more detailed description of the results.
Response 1: In advance, thank you for your feedback/comments. We have moved some of the contents of the Discussion to the Result which is considered still part of the explanation of the results (lines 255-329), besides that we have added supplementary data in the Appendix section to explain the results in detail (lines 449-539).
Point 2: Moreover, there is no discussion of the obtained results against the background of the available literature.
Response 2: In advance, thank you for your feedback/comments. In the Discussion section, we have improved by explaining the performances and limitations of the method (lines 332-349), suggestions for further research (lines 350-359), and some new information/contributions are presented that were not present in the previous oil palm tree counting research (lines 360-366).

Reviewer 2 Report
Dear authors,
In my opinion, this manuscript could be interesting for readers. However, I consider that the manuscript needs major revisions!
The introduction provides sufficient background, the methods and data are sufficiently well described, but I think the results should be developed. Also, the discussions are missing in this form of the manuscript.
In my opinion, the discussion chapter is actually part of the results. Therefore, I suggest moving it or some parts of it to the results chapter. Some of the figures should be moved to supplementary materials and I would recommend add a map with the final results obtained for the entire studied area.
I also recommend reorganization the discussion chapter. It must present the main results of the study (without duplicating them) in correlation with the results of previous studies (supported by relevant references); a comparison between what was obtained in this study and previous findings. Also, I recommend presenting what is new in this study compared to previous studies, possible limitations of the study.
Author Response
Response to Reviewer 2 Comments
Point 1: The introduction provides sufficient background, the methods and data are sufficiently well described, but I think the results should be developed. Also, the discussions are missing in this form of the manuscript. In my opinion, the discussion chapter is actually part of the results. Therefore, I suggest moving it or some parts of it to the results chapter. Some of the figures should be moved to supplementary materials and I would recommend add a map with the final results obtained for the entire studied area.
Response 1: In advance, thank you for your feedback/comments. We have moved some of the contents of the Discussion to the Result which is considered still part of the explanation of the results (lines 255-329), besides that we have added supplementary data in the Appendix section to explain the results in detail (lines 449-539).
Point 2: I also recommend reorganization the discussion chapter. It must present the main results of the study (without duplicating them) in correlation with the results of previous studies (supported by relevant references); a comparison between what was obtained in this study and previous findings. Also, I recommend presenting what is new in this study compared to previous studies, possible limitations of the study.
Response 2: In advance, thank you for your feedback/comments. In the Discussion section, we have moved some of the contents of the Discussion to the Result which is considered still part of the explanation of the results (lines 255-329), and also we have improved by explaining the performances and limitations of the method (lines 332-349), suggestions for further research (lines 350-359), and some new information/contributions are presented that were not present in the previous oil palm tree counting research (lines 360-366).

Round 2
Reviewer 1 Report
All my comments have been considered and the manuscript has been improved.
Reviewer 2 Report
The authors responded to all my comments, and I believe that the manuscript has been sufficiently improved to warrant publication in BDCC.